# The Methylene-Cycloalkylacetate (MCA) Scaffold in Terpenyl Compounds with Potential Pharmacological Activities

**DOI:** 10.3390/molecules24112120

**Published:** 2019-06-05

**Authors:** Ignacio E. Tobal, Alejandro M. Roncero, Rosalina F. Moro, David Díez, Isidro S. Marcos

**Affiliations:** Departamento de Química Orgánica, Facultad de Ciencias Químicas, Universidad de Salamanca. Plaza de los Caídos 1-5, 37008 Salamanca, Spain; ignaciotobal@usal.es (I.E.T.); alexmaron@usal.es (A.M.R.); rfm@usal.es (R.F.M.); ddm@usal.es (D.D.)

**Keywords:** methylene-cycloalkylacetate, MCA, *ent*-halimic acid, halimanes, neurotrophic

## Abstract

Recently, the methylene-cycloakylacetate (MCA) scaffold has been reported as a potential pharmacophore for neurite outgrowth activity. In this work, natural diterpenes that embed MCA fragments are reviewed, as they are major components of *Halimium viscosum*: *ent*-halimic acid, the prototype for these bioactive compounds. Herein, structures, sources, and activities for the natural diterpenes, as well as their synthetic derivatives of interest, are reviewed.

## 1. Halimanes Containing the MCA Fragment

Recently, molecules containing methylene-cycloalkylacetate (MCA) fragments (Figure 1) have been reported to show very interesting neurotrophic properties [1]. The MCA scaffold can be observed in many easily available terpenes of natural origin and synthetic derivatives, many of which show interesting pharmacological activities [2,3,4,5,6].

Although different terpenoids containing the MCA scaffold have been reported, in this work only the natural diterpenoids will be reviewed. Among diterpenoids containing the MCA fragment, halimane is the most structurally diverse and numerous family.

As the MCA moiety can be considered as a pharmacophore for neurotrophic activity, these natural diterpenoids could be understood as novel neurotrophic lead compounds, but show other interesting biological (e.g., antitumor and antifeedant) activities [7,8].

Firstly, halimane diterpenoids, with an MCA fragment embedded, with eventually potential neurotrophic activity and other biological activities are discussed. They have been classified into three groups according to the annular double bond found: in Ring A, Group I (Figure 2), in Ring B, Group II (Figure 3), or, as secoderivatives, in Group III (Figure 4). The compounds of each group have been included in Table 1, Table 2 and Table 3, respectively, where the natural source, the biological activity (if studied), and references are given. As can be observed in Group I (Table 1), nearly all are *ent*-halimanes and degraded compounds such as di-, tri-, and tetranorderivatives. Group II contains halimanes from the normal and enantiomeric series. All the secoderivatives, the Group III compounds, correspond to the enantiomeric series. All compounds have been isolated from plants of different families, and only two of them have been found in microorganisms—bacteria of the genus *Micromonospora.*

### 1.1. Group I: The Halim-1(10)-Enes Group 

This group is composed of 46 *ent*-halimanes **1**–**46** and a halimane **47** (Figure 2). All of them have a carboxylic function at C18, except **40** and **41** where the function appears at C19 as a hydroxyl group or a carbocyclic acid. This group contains compounds such as chettaphanin I, **1**, the first halimane known, and *ent*-halimic acid, characterized as its methyl ester, **2**, from which the group attains its name, and which has been used as a starting material to establish the absolute configuration of chettaphanin I, **1**, and for the synthesis of many *ent*-halimanes, as will be discussed later on.

### 1.2. Group II: The Halim-5-Enes Group

This group contains 30 compounds (Figure 3). Nine of them (**48**–**56**) belong to the enantiomeric series called *ent*-halimanes, two of which are nor-*ent*-halimanes, and the rest of the halimanes belong to the normal series (**57**–**77**), among which six 8-*epi*-halimanes (**72**–**77**) can be found, where three are tetranorderivatives (**75**–**77**).

### 1.3. Group III: Secohalimanes and 4-Enes Derivatives

All secohalimanes (**78**–**89**) that contain the MCA fragment are 3-seco derivatives (Figure 4). All of them contain a furan fragment in the side chain and four present a butan-20,12-olide. The only halim-4-en derivative is a norderivative known as teucvin or mallotucin A, **89**.

## 2. Ent-Halimic Acid as a Precursor of Biologically Active Compounds and Other Derivatives of Interest

*Ent*-halimic acid is very abundant in the *Halimium viscosum* extract, and its methyl ester **2** has been used as a starting material for the synthesis of a series of natural halimanes corroborating their structures, biologically active derivatives and the preparation of other interesting compounds. Figure 5 shows some of the diterpene or sesquiterpene derivatives synthesized from *ent*-halimic acid: 1. *ent*-halimanolides [2,56,57,58]; 2. chettaphanin I and II [4,5]; 3. sesterterpenolides [6,59,60]. Figure 6 shows other groups of compounds prepared from **2**, among which are as follows: 4. hybrid compounds: sesterterpenolide bioconjugates and glycerophospholipids [61]; 5. rearranged derivatives: *ent*-labdanes [62], abeopicrasanes [63], and a propellane [64]; 6. sequiterpene-quinone/hidroquinones [65]; 7. terpene alkaloids: sesqui- and diterpene-alkaloids [7,8,66,67,68,69].

## 3. Synthetic Transformations of *Ent*-halimic Acid Methyl Ester, 2

In this section, the synthesis of these compounds is discussed. The synthetic routes starting from *ent*-halimic acid methyl ester **2** have made it possible to establish or corroborate the structure of different natural *ent*-halimanes and access other interesting derivatives due to their rearranged structures or to their biological activities. Many of the synthesized intermediates possess the MCA fragment in their structures. 

### 3.1. Synthesis of Ent-Halimanolides

The first three natural *ent*-halimanolides **90**, **91** and **92** were synthesized from *ent*-halimic acid using the methyl ketone **94** as intermediate [2,56] (Scheme 1). This methyl ketone **94** was obtained from *ent*-halimic acid methyl ester **2** by a six-step procedure that includes reduction of the methoxycarbonyl at C18. Bestmann methodology was used for the synthesis of butenolide **96** [70], the key intermediate in the synthesis of *ent*-halimanolides **90**, **91**, and **92**. The decalin double bond isomerization of **96** to the tetrasubstituted position permits the synthesis of **90**. The γ-hydroxybutenolides **91** and **92** can be obtained using Boukouvalas methodology [71]. Biological testing has been carried out on these compounds. The compound **96** shows cytotoxic and antiviral activity [HELAM cells (IC_50_ = 5.0), MDCK (IC_50_ = 5.1) and influenza virus (IC_50_ = 6.8)] [56].

Due to the biological (antifeedant, anti-inflammatory, antitumor, and antimicrobial) activity related to diterpenoids and sesterterpenoids containing the hydroxybutenolide scaffold, a synthesis of *ent*-halimanolides, structurally related to natural occurring lactones, was designed, starting from *ent*-halimic acid (α and β-hydroxybutenolides: 15,16-olides **97**–**100** and 16,15-olides **101**–**104**) [58]. The key intermediate, triol **106** (Scheme 2), was obtained by selective photooxygenation and ulterior hydroboration. The selective protection of **106** as 1,3-dioxolane or 1,3-dioxane led to the hydroxyderivatives **107** and **108**, respectively, after oxidation and deprotection of which produced the β-hydroxybutanolides **97**–**100** (in a ratio 27:33:20:20) and the α-hydroxybutanolides **101**–**104** (in a ratio 22:22:28:28), respectively. The relative configuration of these compounds was established by ^1^H NMR, and the absolute configuration achieved by CD spectroscopy. The structures of **100** and **103** were confirmed via X-ray. Lactone **109** was synthesized by a 5-step sequence using methylketone **110** as an intermediate [29]. Using Reformatsky methodology [72,73], β-hydroxybutanolides **109** was obtained. Antifeedant testing has been carried out on these compounds. The α-hydroxybutanolides **103** and **104** are weakly active, and β-hydroxybutanolides **109** are moderately active [58].

In the same manner, an efficient synthesis of the *ent*-halimanolide **112** (15,16-epoxy-12-oxo-*ent*-halima-5(10),13(16),14-trien-18,2β-olide), from *ent*-halimic acid, has been achieved, corroborating the structure of the natural compound and establishing its absolute configuration [57] (Scheme 3).

By using the dinorderivatives **113** and **114** as intermediates, the tetranorderivative **116** was accessed by a new route at the multigram scale. From **116**, by a six-step sequence, the natural lactone **112**, moderately active against HeLa (human cervix cancer), was obtained [57]

### 3.2. Chettaphanin Synthesis

Chettaphanins I, **1**, and II, **123**, isolated from *Adenochlaena siammensis* Ridl (Euphorbiaceae) in 1970 [9] and 1971 [10] are the first two *ent*-halimanes discovered. Both compounds are the main components of “chettaphangki,” a digestive remedy used in folk medicine in Thailand. Their structures have been determined by chemical and spectroscopic correlations. The chettaphanin II structure was corroborated by X-ray crystallography of a chettaphanin II derivative. The absolute configuration that remained undetermined was established by synthesis using *ent*-halimic acid as a starting material [4,5] (Scheme 4). The key intermediate **125** was accessed from *ent*-halimic acid methyl ester **2** by a seven-step sequence, with good global yield. This sequence includes, side chain degradation to the corresponding dinorderivative followed by C2 functionalization and Baeyer–Villiger oxidation until the corresponding tetranorderivative **124** was obtained. The protection of enone **124** was followed by hydrolysis of the acetoxyl group and oxidation, leading to aldehyde **125**. During the enone protection with ethylene glycol, the dioxolane formed, and the isomerization of the double bond to the decalin tetrasubstituted position took place at the same time. Furyllithium addition to **125** followed by oxidation led to intermediate **126**. Chettaphanin II, **123**, was obtained by a simple treatment in acidic media of **126**, and chettaphanin I, **1**, was obtained by epoxidation of **126**, followed by treatment in acidic media. 

### 3.3. Sesterterpenolide Synthesis

Several sesterterpenoids isolated from marine natural sources contain a residue of γhydroxybutenolide as a characteristic structural fragment. This fragment is responsible in many cases for the bioactivity of the corresponding sesterterpenolides.

Cladocorans A and B, isolated from the mediterranean coral^6^
*Cladocora cespitosa* (L) by Fontana et al. [74] (Figure 7), are sesterterpenolides, and their structures, **127** and 1**28**, were established according to their spectroscopic properties. These sesterterpenolides are structural analogues of the natural dysidiolide [75,76], an inhibitor of protein phosphatases cdc25A (IC50 = 9.4 *μ*M) and cdc25B (IC50 = 87 *μ*M), which are essential for cell proliferation. Dysidiolide inhibits the growth of A-549 human lung carcinoma and P388 murine leukaemia cell lines at low micromolar concentrations [77,78,79,80,81]. These important physiological activities of the dysidiolide attract the attention of chemists, biologists, and pharmacologists. Compounds **127** and **128** can be considered as isoprenyl-halimanes and their potential biological activities inspired us to synthesize them with some analogues using the methyl ester of *ent*-halimic acid **2** as a starting material. The synthesis by our group of compounds **127** and **128** and their epimers at C18 (**129** and **130**) demonstrate that the structures proposed by Fontana et al. for cladocorans A and B (**127** and 1**28**) should be revised. The natural product structures for cladocorans A and B were finally revised by Miyaoka and colleagues [3] (Figure 7), and the correct structures of these natural products appear in Figure 7. It was found that cladocoran B is an olefinic regioisomer of dysidiolide, and cladocoran A is its acetate.

The synthesis of bioactive sesterterpenoid γ-hydroxybutenolides 15,18-bisepi-*ent*-cladocoran A and B, 1**27** and 1**28** (Scheme 5), and the epimers of these compounds at C18, 15-*epi*-*ent*-cladocoran A and B, **129** and **130**, using *ent*-halimic acid methyl ester **2** as a starting material was achieved (Figure 7). Starting from *ent*-halimic acid methyl ester **2**, the key intermediate **131** was accessed by a degradation sequence of the side chain of four carbon atoms and elongation of C18 by introduction of the south chain. The furosesterterpenoid **132** was obtained by introducing the furan fragment by the addition of furyllithium, and the isoprenic unit of the south chain was completed by coupling the adequate Grignard reagent with the iododerivative or the tosylderivative of **131**. The corresponding epimers at C18 of **132** were separated by column chromatography. In each of them, the γ-hydroxybutenolide unit was finally obtained using Faulkner methodology [82], obtaining in each case **127**, **128**, **129**, and **130**. The synthesized sesterterpenolides **127**, **128**, **129**, and **130** inhibited cellular proliferation (IC_50_ ≈ 2 µM) of a number of human leukaemic and solid tumor cell lines [60].

The promising biological activities showed that, in some cases, sesterterpenolides **127**, **128**, **129**, and **130**, dysidiolide analogues, are more active than the compound of reference dysidiolide and boost the search for new analogues. In this manner, several sesterterpenolide analogues of dysidiolides **135**–**139** (Scheme 6) have been synthesized from *ent*-halimic acid methyl ester **2**, according to Scheme 6 [59]. The main structural change with the previous cladocoran derivatives is the situation of the γ-hydroxybutenolide in the south side chain of the molecule. The antitumoral activity in vitro against human HeLa, A549, HT-29, and HL-60 carcinoma cells was achieved. The proliferation inhibition data showed significant antitumor activity in the compounds **135**–**139**, inhibiting proliferation of distinct cancer cell types with an IC_50_ in the low micromolar range (Scheme 6) [59].

### 3.4. Synthesis of Hybrid Compounds of Sesterterpenolides with Glycerols

One of the sesterterpenolides, **135**, obtained from *ent*-halimic acid methyl ester **2** was used in the synthesis of a series of bioconjugated sesterterpenoid with phospholipids and polyunsaturated fatty acids (PUFAs) such as **141** and **142** (Figure 8), potentially bioactive compounds and antiproliferative agents in several cancer cell lines [61]. Each of the compounds that participate in these bioconjugates show antitumor activities, and a synergistic effect was expected to result from their conjugation [83,84,85]. Different substituted analogues of the original lipidic ether edelfosine [86] (1-*O*-octadecyl-2-*O*-methyl-rac-glycero-3-phosphocholine) were obtained while varying the sesterterpenoid in position 1 or 2 of the glycerol or a phosphocholine or PUFA unit in position 3. Simple bioconjugates of sesterterpenoids and eicosapentaenoic acid (EPA) have also been obtained. All synthetic derivatives were tested against human tumor cell lines HeLa (cervix) and MCF-7 (breast). Some compounds showed good IC_50_ (0.3 and 0.2 µM) values against these cell lines [61].

### 3.5. Synthesis of Rearranged Compounds

Due to its functionality, *ent*-halimic acid methyl ester **2** can be considered as an excellent synthon for the synthesis of new natural products and as a starting material in the synthesis of rearranged compounds and compounds of interest in perfumery.

#### 3.5.1. Synthesis of *Ent*-Labdanes from *Ent*-Halimanes.

Scheme 7 shows the 1,2-shift of Me-20 of *ent*-halimanes (**143**) that leads to *ent*-labdanes (**144**). This rearrangement will be the opposite to the biosynthetic route in which the *ent*-labdanes are the precursors of *ent*-halimanes. For the first time, *ent*-labdanes have been synthesized starting from *ent*-halimic acid methyl ester **2**, following a route that is the reverse of the biosynthetic one leading to the former compounds [62] (Scheme 7). Effectively, the *ent*-halimane epoxyderivative **143**, formed in three steps from **2**, led to the *ent*-labdane tetranorderivative **144** by treatment with Lewis acid. The rearrangement allowed the 1(10)-halimanes to be used as starting materials for the synthesis of bioactive labdanes, so this rearrangement could be used for the transformation of compounds in picrasanes with an abeopicrasane skeleton obtained from *ent*-halimic methyl ester **2**, as will be described in the following. 

#### 3.5.2. Synthesis of Abeopicrasanes from *ent*-Halimanes.

Picrasanes are quassinoids and degraded triterpenes and are interesting for their antitumoral properties, bruceantin being one of the best known [87]. An advanced intermediate **147** (Scheme 8) with the ABC ring of the picrasane quassinoid skeleton **148** has been synthesized from *ent*-halimic acid methyl ester **2**. The rearrangement from *ent*-halimanes to *ent*-labdanes previously described (Scheme 7) was thought to apply to the transformation of abeopicrasanes into picrasanes [63]. The bicyclic system of the starting material, *ent*-halimic acid methyl ester **2**, has been incorporated as the BC part of the ABC system. To date, no diterpenes of the antipode series have been used in this kind of approach for the quassinoids with a picrasane skeleton. The tricyclic system was elaborated using the tetranorderivative **145** as an intermediate, performing the allylic oxidation and incorporating on C18 the four carbon atoms necessary to access the 4,5-secoabeopicrasane **146**. The quaternary carbon C-4 was incorporated as C-10, and carbon C-5 incorporated as C-9, with the right configuration. Annulation of dione **146** led to the adequate abeopicrasane **147**, exploiting the rearrangement described before. 

#### 3.5.3. Synthesis of [4.3.3] Propellanes from *Ent*-Halimanes.

*Ent*-halimic acid methyl ester **2** was used as the starting material for an efficient synthesis of a series of tetranorderivatives **149**–**151**, functionalized at C-18 (Scheme 9). These compounds could be used to synthesize the [4.3.3] propellane **154**, the oxide **152**, and the lactone **153** similar to ambreinolide, all of which may be of interest to those in the perfume industry [64].

### 3.6. Quinone/Hydroquinone Sesquiterpenes

The quinone/hydroquinone sesquiterpenes are compounds mainly of marine origin and are interesting for their structural variety and biological activities [88]. *Ent*-halimic acid methyl ester **2** has been used in the synthesis of the quinone/hydroquinone sesquiterpene (-)-aureol **155**, the (-)-smenoqualone **157**, and the (-)-neomamanuthaquinone **156** and in the formal synthesis of the (-)-cyclosmenospongine **158** (Scheme 10) with, for example, antiinflammatory, antimicrobial, antitumor, or antiviral activities.

The syntheses of those quinone/hydroquinone sesquiterpenes **155**–**158** from *ent*-halimic acid methyl ester **2** were planned according to the synthetic sequence in Scheme 10, which was developed by the AB/ABD/ABCD approach. Effectively starting from *ent*-halimic acid methyl ester **2**, the tetranoderivative intermediate **159** was prepared, and from this, key intermediate **160,** which incorporates Ring D, was synthesized, which included a Barton decarboxylation reaction in the presence of benzoquinone [89]. The reduction of **160** and ulterior cyclization of hidroquinone **161** yielded (-)-aureol **154**. The cyclization to obtain the tetracyclic compound was achieved with *p*-TsOH and in a stereoselective manner with F_3_B·Et_2_O. With **160** in hand, (-)-neomamanuthaquinone **156** (Scheme 10) can be obtained by the addition of NaOMe to the quinone ring, according to the procedure of Theodorakis and co-workers for the synthesis of ilimaquinone [90]. This tricyclic sesquiterpene quinone/hydroquinone, **156**, give access to (-)-smenoqualone **157** and (-)-cyclosmenospongine **158**.

### 3.7. Sesqui- and Diterpene-Alkaloids

*Ent*-halimic methyl ester **2** has also been used in the synthesis of terpene-alkaloids, in particular in the preparation of 7,9-dialkylpurines ((+)-agelasine C) and other diterpene- and sesquiterpene-indoles.

#### 3.7.1. Synthesis of Diterpene-alkaloid, (+)-agelasine C [7]

Agelasines are a family of diterpene-alkaloids isolated from marine sponges of genus Agelas [91]. Agelasine C is one of the first four known agelasines, shows powerful inhibitory effects on Na and K-ATPase, and shows antimicrobial activities (Figure 9). Nakamura et al. in 1994 presented the structural formula **162** for agelasine C determined by spectroscopic methods [92]. *Epi*-agelasine C, **163**, is an antifouling substance active against macroalgae and was isolated in 1997 by Hattori et al. [93] from the marine sponge *Agelas mauritiana*, whose structure was proposed by spectroscopic methods (Figure 9). 

The interest in *epi*-agelasine C as an antifouling agent [93] and the necessity of establishing the absolute configuration led to the decision to synthesize **164** (Scheme 11). This synthesis was carried out according the synthetic strategy shown in Scheme 11, analogue to the ones developed for other agelasines, consisting in coupling the adequate terpenic fragment (**165**) with a purine (**166**) [7].

The physical properties of the synthesized product **164** are very different from the natural product *epi*-agelasine C, whose proposed structure of **163** (Figure 9) should be revised. On the contrary, comparing the ^13^C NMR spectra of **164** with agelasine C, with the proposed structure of **162**, showed that they were identical to their ^1^H NMR spectra. Although the specific rotation for **164** and agelasine C have a similar absolute value, they have a different sign. It should be concluded that the structure for (-)-agelasine C should be corrected to the structure of **165**, an enantiomer of the synthetic product **164** (+)-agelasine C (Figure 10).

Spectroscopic considerations that arose by comparison of the spectroscopic data of **164** with those of *epi*-agelasine C and by the specific rotation of both compounds suggested the structure that appears in Figure 10 [7] for the natural product *epi*-agelasine C **166**.

#### 3.7.2. Synthesis of the Indole Diterpene-alkaloid (+)-thiersindole C

The indole diterpene-alkaloid (+)-thiersindole C **167** was synthesized from *ent*-halimic acid methyl ester **2** [8] (Scheme 12). Firstly, the bicyclic system was elaborated by the adequate functionalization of C3, preparing intermediates **168** and **169**. Secondly, a Fischer indolization was used in order to obtain the north side chain in intermediate **170**, and the elongation of the south side chain with an isoprene unit was finally afforded in two steps through **171**, leading to (+)-thiersindole C **167**. The synthesis of (+)-thiersindole C **167** corroborated the absolute configuration of the natural product (-)-thiersindole C. The synthesized (+)-thiersindole C **167** showed antitumor activity against a number of several human tumor cell lines with an IC_50_ in the range of 10^−5^ M [8].

#### 3.7.3. Synthesis of Sesquiterpenil-Indoles

The synthesis of 12-*epi*-*ent*-polyalthenol **172** and 12-*epi*-*ent*-pentacyclindole **174** from *ent*-halimic acid methyl ester **2** as starting material was carried out using the following as intermediates: the trinorderivative **175** and the trinorderivatives functionalized in C3 **176**-**177** [66,67] (Scheme 13). The synthesis of the pentacyclic compounds can be considered biomimetic, as our group performed a cyclization of 3-(but-3-enyl) indole derivatives that produce polycyclic compounds with a hexahydrocarbazole structure. In this reaction, three stereogenic centers are generated in one step [69]. The structure of the natural product polyalthenol **178** and pentacyclindole **179** (Figure 11) were confirmed in this way. 

Several of the sesquiterpene-indoles synthesized show cellular proliferation inhibition of a number of human leukaemic and solid tumor cell lines [66,67].

A series of sesquiterpene-indole (**180**–**203** and **204**–**221**) analogues of polyalthenol **178** and pentacyclindole **179**, respectively, were synthesized (Figure 12) starting from *ent*-halimic acid methyl ester **2** in order to test their biological activity [68]. These 42 analogues include diverse oxidation levels at the sesquiterpenyl moiety and different functionalization on the indole ring. All derivatives were tested against a representative panel of Gram-positive and Gram-negative bacterial strains, and the human solid tumor cell lines A549 (non-small cell lung), HBL-100 (breast), HeLa (cervix), SW1573 (non-small cell lung), T-47D (breast), and WiDr (colon). Overall, the compounds presented activity against the cancer cell lines. The resulting lead, displaying a polyalthenol scaffold, showed GI50 values in the range 1.2–5.7 µM against all cell lines tested [68].

## 4. Conclusions

Among natural terpenoids, bicyclic diterpenes with a halimane skeleton constitute the family that most often shows the methylene-cycloalkylacetate (MCA) fragment with potential application as neurotrophic agents. *Ent*-halimic acid, a major constituent of *Halimium viscosum*, is the prototype for this MCA-containing diterpene, which has been proven to be an excellent starting material for the synthesis of a variety of biological active compounds and therefore for potentially neurotrophic diterpenes.

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
