# Peer review of "The Methylene-Cycloalkylacetate (MCA) Scaffold in Terpenyl Compounds with Potential Pharmacological Activities"

_molecules, 2019, doi:10.3390/molecules24112120_

Round 1

Reviewer 1 Report

This manuscript describes the MCA scaffold.  It is well reviewed about it.  Therefore, I think this manuscript is acceptable as a Review in Molecules.  However, there are tons of typo.  The authors should check the manuscript carefully.  I write some of them which I noticed.

P. 11, l. 113.  Is TBDMSTf right?

P. 14, l. 195.  Change the name of Cladocoran to small character.

P. 15, l. 227.  Change the style of O to itaclic.

P. 15, l. 234.  Is sterterpenoids right?

All ent should be written in italic.

Author Response

Reviewer 1:

This manuscript describes the MCA scaffold.  It is well reviewed about it.  Therefore, I think this manuscript is acceptable as a Review in Molecules.  However, there are tons of typo. We have changed all detected ones, sorry about that.  The authors should check the manuscript carefully.  I write some of them which I noticed.

P. 11, l. 113.  Is TBDMSTf right? It is correct. Now page 12, 143

P. 14, l. 195.  Change the name of Cladocoran to small character. Done, now page 15 ,227

P. 15, l. 227.  Change the style of O to italic.. Done, now page 17, 273

P. 15, l. 234.  Is sterterpenoids right? Corrected to sestertepenoids. Now page 17, 282.

All ent should be written in italic. Done

Reviewer 2 Report

This review described some compounds with methylene-cycloakylacetate scaffold. This is a big family of natural products. I suggest the authors adding some detailed description about the activities in tables 1-3, just because activity is an very important part for natural products research. In addition, there is a lack of details in the synthesis part. The title of section 1 (Introduction) is not proper enough, so I suggest changing into a better one.

Author Response

This review described some compounds with methylene-cycloakylacetate scaffold. This is a big family of natural products. I suggest the authors adding some detailed description about the activities in tables 1-3, just because activity is an very important part for natural products research. The activities have been included in the tables as suggested. In addition, there is a lack of details in the synthesis part. We have considered enough description of the synthesis in order not to increase the length of the review. Researcher interested could find the complete description in the references that we have included. The title of section 1 (Introduction) is not proper enough, so I suggest changing into a better one.

We have changed it into Halimanes containing the MCA fragment